# Dietary Practices and Anthropometric Status of the Rural University Students in Limpopo Province, South Africa

**DOI:** 10.3390/ijerph22060936

**Published:** 2025-06-13

**Authors:** Lindelani F. Mushaphi, Khutso Mokoena, Anzani Mugware, Alphonce Bere, Selekane Ananias Motadi

**Affiliations:** 1Department of Nutrition, Health Sciences, University of Venda, Private Bag x5050, Thohoyandou 0950, South Africa; lindelani.mushaphi@univen.ac.za (L.F.M.); selekane.motadi@univen.ac.za (S.A.M.); 2Department of Public Health, Health Science, University of Venda, Thohoyandou 0950, South Africa; khutso.acolyte@gmail.com; 3Department of Mathematical and Computational Sciences, Faculty of Science, Engineering and Agriculture, University of Venda, Thohoyandou 0950, South Africa; alphonce.bere@univen.ac.za

**Keywords:** dietary practices, anthropometric status, obesity

## Abstract

Obesity among adolescents has increased over the past decades in most parts of the world in low- and middle-income countries. This study aimed to investigate the dietary practices and anthropometric status of the rural university students in Limpopo Province. A cross-sectional study was conducted among university students in Limpopo Province. A total of 363 students aged 18 to 42 years residing at the university residences were randomly selected. A structured self-administered questionnaire was used to collect data. Anthropometric measurements were conducted by a qualified biokinetist. Data were analysed using SPSS version 29. More than half (57.3%) of the students were females. About 42.1% reported that they consume fast food once per week. The prevalences of overweight and obesity were 21.8% and 7.5%, respectively. A chi-square analysis revealed a significant association between soft drinks, juice, and energised drink consumption frequency and BMI status (*p* = 0.006). Results revealed a significant association between the frequency of eating processed food per week and the WHR status (*p* = 0.013). Overweight and consuming fast food and sugary drinks are significant concerns amongst rural-based university students. The current study recommends that regular nutrition education campaigns be conducted at the university to encourage students to make healthier eating choices.

## 1. Introduction

Obesity among adolescents has increased over the past decades in most parts of the world in low- and middle-income countries, particularly in urban areas [1]. Globally, the WHO estimated that 39% of adults over the age of 18 are overweight and 13% are obese [1]. Between 1975 and 2016, the global prevalence of obesity nearly tripled. Obesity was formerly thought to be a problem in high-income countries, but data from the last decade show that low- and middle-income countries, particularly in Sub-Saharan Africa, carry the double burden of being underweight and overweight [2,3]. South Africa is no exception, as the number of people who are overweight or obese is on the rise [4]. According to Shisana et al. [5], about 25% and 40.1% of females aged 15 years and older are overweight and obese, respectively. In addition, 19.6% and 11.6% of males in South Africa were overweight and obese, respectively [5].

Obesity is a complex condition that results from various factors such as physiological, environmental, and behavioural influences, which all contribute to a sustained positive energy balance [6]. Overweight or obesity increases the risk of developing non-communicable diseases (NCDs) such as type 2 diabetes, mellitus, cardiovascular disease, hypertension, dyslipidaemia, and metabolic syndrome [7]. The non-communicable diseases affect people of all age groups, regions, and countries, and they are the leading cause of mortality. The burden of NCDs such as cardiovascular diseases, cancer, diabetes mellitus, and hypertension are the major causes of death [8]. It is estimated that 74% of global mortality is related to NCDs, and of all these mortalities, 77% occur in low- to middle-income countries [9]. The development of NCDs is associated with poor eating habits, physical inactivity, exposure to tobacco smoke, and excessive use of alcohol [10]. The WHO earlier identified a sedentary lifestyle as one of the leading causes of mortality and morbidity worldwide [11].

Gouda et al. [12] indicated that sub-Saharan Africa is in a rapid epidemiological transition where NCDs are increasing. Students are transitioning from secondary school to university, which involves significant changes in lifestyle behaviour that may influence their long-term health (eating habits) [13]. The lifestyle and eating habits of university students are influenced by factors such as culture, learning about family life and starting to have experiences with others, and living on their own, as well as new discoveries, increased academic responsibilities, and lack of time to prepare and eat meals in an appropriate manner, leading to consumption of practical and convenience snacks which are of low nutritive value [14]. Physical inactivity and the consumption of a diet high in saturated fats and low in fibre contribute significantly to the high prevalence of overweight and obesity [15]. These factors may contribute to body weight, which may enhance the risk of non-communicable disease and changing how people perceive their body image [16].

The university is based in the Vhembe district, which was chosen because of its large modern shopping centres that feature well-known retail chains and take-away food vendors. The availability of large supermarkets and take-away food retailers in the vicinity of the university is likely to promote the consumption of fast food by the students [17,18]. To our knowledge, no published research has been done on the dietary practices and anthropometric status of university students in rural Limpopo Province, South Africa. This represents a critical gap in the literature, as this population may face unique nutritional and lifestyle challenges influenced by their geographic, socioeconomic, and academic environment. While numerous studies have explored similar objectives in other regions or among different populations, the absence of localised data limits our ability to develop targeted interventions or inform context-specific health policies [19,20]. Therefore, this study aims to address this gap by providing relevant insights into the health and nutritional status of university students in this under-researched area. The study, grounded in the 1990 UNICEF Conceptual Framework [21], explores the dietary practices and anthropometric status of rural university students in Limpopo Province, addressing both undernutrition and overnutrition.

## 2. Materials and Methods

### 2.1. Study Design and Setting

A cross-sectional quantitative study was conducted among rural university students in Limpopo Province, South Africa, from June to October 2019. The study aimed to assess the dietary practices and anthropometric status of these students. The university is in Thohoyandou town within the scenic Vhembe District of Limpopo Province. The university is organised into four faculties: Health Sciences; Humanities, Social Science, and Education; Management, Commerce, and Law; and Science, Engineering, and Agriculture. This study was conducted after obtaining ethical clearance from the University Research Ethics Committee. This study was reviewed and approved by the Human and Clinical Trials Research Ethics Committee and the University Research Ethics Committee (SHS/19/12/0605).

### 2.2. Study Population, Sample Size, and Sampling Procedure

The sample size was calculated using Slovin’s formula, based on a population of 16,000 students residing in university residences. A tolerable error level of 0.05 and a 95% confidence level were considered, yielding a required sample size of 363 participants. To account for potential attrition, the sample size was increased by 10%. A list of university residences was obtained from the university headmaster. A random sampling technique was used to select 7 out of 16 university residences, and a convenience sampling technique was employed to select the students. Only students who were present and consented on the day of data collection were included in the study. Students with physical limitations that prevented them from standing independently were excluded due to lack of equipment to measure their height.

### 2.3. Variables Measured

Variables measured were socio-demographic characteristics, anthropometric measurements (weight, height, waist circumference), and dietary practices.

### 2.4. Socio-Demographic Characteristics of Study Participants

The socio-demographic characteristics of the students were estimated via questionnaires given as part of an oral interview. The socio-demographic questionnaire collected data on variables such as age, gender, level of study, schools, income, and source of income.

### 2.5. Anthropometrics Measurements

The anthropometric measurements were conducted in line with the standard procedures established by the International Society for the Advancement of Kinanthropometry [22].

A qualified Biokinetist performed anthropometric measurements. Anthropometric measurements (weight and height) were collected in duplicate, using calibrated equipment with the students wearing light clothing without shoes. Height was measured to the nearest 0.1 cm, via a portable stadiometer; weight was determined to the nearest 0.01 kg on a portable Seca solar scale (model 0213) (Seca, Hammer Steindamm, Hamburg, Germany); and waist circumference was measured using a non-stretchable tape.

Anthropometric indicators were categorised as underweight, normal weight, overweight, and obese. According to WHO guidelines [22], a body mass index (BMI) of less than 18 kg/m^2^ is classified as underweight, 25–30 kg/m^2^ as overweight, and 30 kg/m^2^ or greater as obese. Central obesity was defined using waist circumference cut-off points: ≥88 cm for females and ≥102 cm for males [23]. The waist-to-hip ratio was calculated by dividing the waist circumference by the hip circumference. Values > 0.8 for women or >1.0 for men are regarded as indicative of central obesity [24].

### 2.6. Dietary Practices

The questionnaire was the primary tool used for data collection. Prior to the main data collection phase, it was pretested and piloted among a small group of university students from a similar background to ensure its clarity, cultural relevance, and appropriateness for the target population. The feedback obtained was used to refine the instrument, enhancing its content validity and ensuring that the questions were understandable and suitable for the context. Data were collected on the number of meals consumed each day, meal patterns, snacking behaviours, meal sources, and the weekly frequency of consuming based on following nine food groups: (1) grains, whites, and tubers; (2) vitamin A-rich fruits and vegetables; (3) other fruit and vegetables; (4) legumes and legume products; (5) fats and oils; (6) meat, poultry, and fish; (7) milk and milk products; (8) eggs; (9) other, sweet and salty foods recommended by the World Health Organisation (WHO) [25]. Students were asked to indicate their frequency of consumption by choosing from the following options: ‘daily,’ ‘three times a week or less,’ ‘more than three times a week,’ and ‘never.’ The WHO guidelines were used to evaluate the adequacy of the number of meals consumed per day.

### 2.7. Statistical Analysis

The Statistical Package for the Social Sciences (SPSS for Windows version 29, SPSS Inc., Chicago, IL, USA) was used to analyse quantitative data. The nutritional status of the students was categorised by using appropriate cutoffs for classification. To determine the association between students’ dietary practices and anthropometric status, chi-square (χ^2^) tests and Kruskal–Wallis tests were used. A *p* < 0.05 was considered statistically significant.

## 3. Results

### 3.1. Socio-Demographic Characteristics of Students

More than half of the students who participated in the study were females. The mean age of the students was 21.8 ± 2.57 years. Of the 363 university students, most (85.7%) were undergraduate. About 18.5% of the students were from the school of education, while 17.4% were from management and 16% were from mathematics and natural sciences. The faculties with low participants were health sciences (10.7%), agricultural sciences (6.9%), and law (4.7%). The majority (81%) of the students were bursary holders, with very few (9.9%) receiving allowances from parents/guardians (Table 1).

### 3.2. Dietary Practices of Students

Table 2 presents the weekly consumption frequency of various food items among students. Approximately 42.1% of students reported eating fast food once a week. In contrast, only 20.7% of students consumed fruits and vegetables on a daily basis. Within the protein food groups, around one-third of participants consumed fish, chicken, eggs, or lean meat four times or more per week. Additionally, about 41% of students ate beans once a week. Lastly, one-quarter of students consumed milk, yogurt, or similar dairy products once per week.

### 3.3. Anthropometric Status of Students

The prevalences of underweight, overweight, and obesity were 4%, 21.8%, and 7.5%, respectively. Overweight rates were higher among females at 22.9% compared to 20.6% among males. Additionally, 11% of female participants were obese, compared to 3.1% of male participants (Figure 1).

The body mass index cut-off point of <18 kg/m^2^ is classified as underweight, 25–30 kg/m^2^ is classified as overweight, while ≥30 kg/m^2^ is classified as obesity.

### 3.4. The Relationship Between Dietary Practices and Body Mass Index (BMI) Status of Students

Table 3 demonstrates the relationship between dietary practices and the body mass index (BMI) status of the students. A chi-square analysis revealed a significant association between soft drinks, juice, and energised drink consumption frequency and BMI classification (*p* = 0.006). Two-thirds (64.9%) of students who were drinking soft drinks, juice, and energised drinks had normal BMI status, while 24.5% were overweight/obese. The finding also showed that non-consumers had the highest prevalence of overweight/obesity (57.7%), suggesting potentially reverse causality (overweight individuals reducing intake) or confounding lifestyle factors.

Eleven percent (11%) of the students were found to have central obesity. The prevalence was notably higher among female students, with 16% affected, compared to just 3% of male students (Figure 2).

### 3.5. The Association Between Dietary Practices and Waist-to-Hip Ratio Status of Students

A chi-square analysis revealed a statistically significant association between the frequency of restaurants per week and (WHR) risk categories (*p* = 0.038). Students who reported never eating out (‘not at all’) showed the most protective profile, with 99.1% falling in the normal WHR category (standardised residual = +2.38) and only 0.9% in the at-risk group (standardised residual = −2.38). There was a statistically significant association between the frequency of eating processed food per week and the WHR status of university students (*p* = 0.013). The results found between bean consumption frequency per week and (WHR) risk categories, with moderate intake showing the most favourable outcomes (*p* = 0.000) (Table 4).

### 3.6. The Association Between University Students’ Anthropometric Status and Dietary Practices

This study also examined the relationship between various dietary consumption frequencies per week and the anthropometric status amongst the university students using non-parametric Kruskal–Wallis tests. The results revealed that none of the dietary factors showed statistically significant associations with the BMI status of the university students after multiple testing corrections (all *p* > 0.05).

Table 5 shows that the median WHR values significantly differed by frequency of eating in restaurants (χ^2^ = 15.54, *p* = 0.001), consumption of protein-rich foods (χ^2^ = 12.23, *p* = 0.016), and consumption of beans/legumes (χ^2^ = 16.58, *p* = 0.002). Students who ate at restaurants twice weekly had the highest median WHR (0.80), suggesting a potential link between frequent dining out and central adiposity. Similarly, daily consumers of beans/legumes had a notably higher WHR (0.83) compared to less frequent consumers, while those consuming protein foods three or more times per week also had elevated WHR values. No significant differences in WHR were observed for intake of fruit, processed foods, dairy products, or sweetened beverages, or number of meals per day.

## 4. Discussion

In the rural university of Limpopo, overweight and fast food consumption are problems. The study findings suggest that the frequency of fast food consumption is linked to both overweight and central obesity. The current findings of a high prevalence of overweight and fast food consumption are similar to those reported at Trayka University [26] and Saudi Arabia University [27]. In the present investigation, overweight and obesity rates were higher among females compared to their male counterparts. The high prevalence of overweight and obesity among females may be explained by the fact that more females eat fast food. Being overweight or obese is regarded as a sign of affluence, riches, and good health for women in South Africa [28]. In African communities, being overweight or obese has a lot of positive connotations [29]. Technology has changed students’ lifestyles, changing how they eat and exercise and increasing their risk of disease [28,30]. The high frequency of waist circumferences greater than the normal value of 88 cm for women is consistent with the high prevalence of overweight and obesity, as shown by BMI values.

Fast food consumption has reached its peak in many middle-income countries, including South Africa. In the current study, over two-thirds of the students reported eating fast food once a week. The results of this study are consistent with reports from Al-Otaibi and Basuny [27] in Saudi Arabian universities and Onurluba and Yilmaz [26] in Trayka University. A study conducted in Limpopo Province revealed that fast food consumption is a more dominant dietary pattern amongst young adults [31]. Fast food consumption remains popular among young people, despite clear evidence of the negative impact on their health [32]. Eating fast food raises the risk of having a poor-quality diet, consuming more calories and fat, and having a diet that is low in micronutrients [33]. Frequent consumption of fast foods may contribute to weight gain, obesity, type 2 diabetes, and coronary artery disease [33].

More than half of the students in the current survey reported that they consume soft drinks, juice, or energy drinks three to six times each week. This was despite the South Africa Food-Based Dietary Guidelines (SAFBDGs) recommending consuming sugar-containing foods and beverages in moderation to reduce the risk of acquiring cardiovascular illnesses, obesity, and type 2 diabetes mellitus, which can lead to premature death [34]. The findings of the current study are in line with the SADHS reports of 2016. High sugar intake increases the risk of chronic disease through weight gain and the emergence of risk factors brought on by a negative glycaemic response [6]. High sugar intake increases the risk of chronic disease through weight gain and the emergence of risk factors brought on by a negative glycaemic response [6].

Three-quarters of students in the current study reported that they usually consume processed meats such as boerewors, cold meats, and mincemeat. Of those who consume processed meat, one-quarter usually consume processed meat at least three to six times per week. Daily consumption of processed food was observed in over half of the college student population in Indiana, USA [35]. These foods are high in saturated fats, which are linked to a greater risk of non-communicable diseases, obesity, hypertension, high cholesterol, and premature death [36]. The findings of the current study strongly indicate that most students consume Westernised diets, rich in energy-dense, processed, and convenience foods high in sugar and fat but low in essential micronutrients. In response to the country’s escalating obesity and NCD burdens costs, policies for salt reduction (2016) and sugar-sweetened beverage taxation (2017) were enacted more recently [37].

This study’s interesting finding was that one-third of students did not consume fruits and vegetables. This occurred even though South African Food-Based Dietary Guidelines (SAFBDGs) emphasises fruit and vegetable consumption on daily basis for people seven years and older [38]. University students often consume insufficient amounts of fruits and vegetables, according to past studies [39,40,41]. A high intake of fruits and vegetables is a cornerstone of a healthy diet and is crucial in reducing the risk of cardiovascular disease [42]. Inadequate intake of fruits and vegetables is linked to elevated risk of overweight, non-communicable diseases, and deficiencies in essential micronutrients [38,43]. The findings are consistent with Bede et al. [39], who reported that less than half of the students at the university consumed fruits and vegetables.

The results of the chi-square analysis revealed significant associations between both the frequency of eating processed foods and the frequency of eating in restaurants per week with the waist-to-hip ratio (WHR) status of university students. The significant association between processed food consumption and WHR is in line with a previous study that indicated diets high in processed foods, which are typically rich in unhealthy fats, refined sugars, and sodium, contribute to increased abdominal fat and higher WHR [44,45]. Processed foods, such as snacks, sugary drinks, and pre-packaged meals, tend to be calorie-dense and low in essential nutrients, leading to an energy imbalance and subsequent weight gain, particularly around the abdomen [46]. Consuming these foods may alter the body’s metabolic processes, contributing to central adiposity, which is reflected in a higher WHR.

Participants who reported eating out twice or more per week exhibited significantly higher median WHR values (0.80) compared to those who ate out less frequently or not at all. This finding is consistent with studies by Lachat et al. [44] and Nago et al. [47], which reported that frequent consumption of meals prepared outside the home, particularly fast food, is associated with poor dietary quality and increased adiposity. Restaurant meals are often energy-dense, high in fat and sugar, and typically served in large portion sizes. When the excess energy from these meals is not expended through physical activity, it can contribute to the accumulation of abdominal fat, thereby increasing the waist-to-hip ratio [48].

Interestingly, participants who consumed protein foods three or more times per week also had significantly higher median (WHR) values (0.79) than those consuming them once per week (0.75). While protein intake is generally associated with improved satiety and weight management, the type and quality of protein may mediate its effects on body composition. Diets high in red or processed meats have been linked to increased visceral fat and metabolic disorders, suggesting that not all protein sources contribute equally to health outcomes [49]. The results may reflect a pattern of high intake of less healthy protein sources, potentially contributing to a higher WHR.

In contrast, participants who consumed beans and legumes daily recorded the highest WHR values (0.83), which were significantly greater than those observed in all other consumption frequency groups. Beans and legumes are typically rich in fibre and plant-based protein, which have been associated with reduced central obesity and improved metabolic health [50]. However, this unexpected finding may be explained by reverse causality, wherein individuals with higher WHR may be increasing legume intake as part of dietary changes aimed at improving health [51]. Additionally, cultural or regional dietary patterns might influence how beans are prepared and consumed, for instance, in high-calorie stews or with added fats, which may diminish their health benefits [45]. These results emphasise the complex relationship between diet and body fat distribution. While certain eating behaviours, such as frequent eating out, are consistently linked to higher WHR [44], the effects of specific food groups like protein and legumes depend heavily on preparation methods, portion sizes, and the overall dietary context. Future research should consider longitudinal data to clarify causality and include detailed dietary assessments to distinguish between food quality and source.

The results of the study revealed significant associations between both the frequency of eating processed foods and the frequency of eating in restaurants per week with the waist-to-hip ratio (WHR) status of university students. The significant association between processed food consumption and WHR is in line with a previous study that indicated diets high in processed foods, which are typically rich in unhealthy fats, refined sugars, and sodium, contribute to increased abdominal fat and higher WHR [44,45]. Processed foods, such as snacks, sugary drinks, and pre-packaged meals, tend to be calorie-dense and low in essential nutrients, leading to an energy imbalance and subsequent weight gain, particularly around the abdomen [46,51]. Consuming these foods may alter the body’s metabolic processes, contributing to central adiposity, which is reflected in a higher WHR.

This study was conducted in one university of Limpopo Province. The results of this study cannot be generalised to all university students in South Africa; however, similar trends are likely to occur in a comparable rural environment. The strength of the study is that it contributes to the existing literature on anthropometric and dietary practices of students at rural universities. It provides a valuable insight on the relationship between dietary practices, BMI, and central obesity among rural-based universities.

### Limitations and Strengths

A limitation of the study is the use of a simplified food group classification in the questionnaire, which restricts the ability to accurately assess dietary adequacy and quality. Specifically, grouping diverse protein sources such as fish, chicken, eggs, and lean meat into a single category overlooks important differences in their nutritional profiles. This lack of specificity may lead to an underestimation or overestimation of dietary diversity and nutrient intake, thereby limiting the precision of conclusions drawn about the healthfulness of the participants’ diets. Despite these limitations, the study provides valuable insights into dietary practices and anthropometric status of the rural university students in Limpopo Province, South Africa, which can inform strategic interventions and public health policies.

## 5. Conclusions

Overweight, consumption of fast food and sugary drink, and skipping of meals are a significant concern in the rural University of Limpopo. The levels of overweight and consumption of fast food and sugary drinks may be attributed to two factors: market liberalisation, which makes fast food more accessible to consumers as a result of market globalisation, and dietary shifts that have been seen in South Africa, from low-fat diets to typical Westernised high-fat diets. In the current study, consuming fast food, drinking sugary drinks, and eating more than three times per day were significantly associated with higher BMI and waist circumference among university students. The higher rate of overweight among females in South Africa than among males may be due to the belief that being overweight or obese is a sign of excellent health, wealth, contentment, and success for females. The current study recommends that regular nutrition education campaigns be conducted at the university to encourage students to make healthier eating choices.

## Figures and Tables

**Figure 1 ijerph-22-00936-f001:**
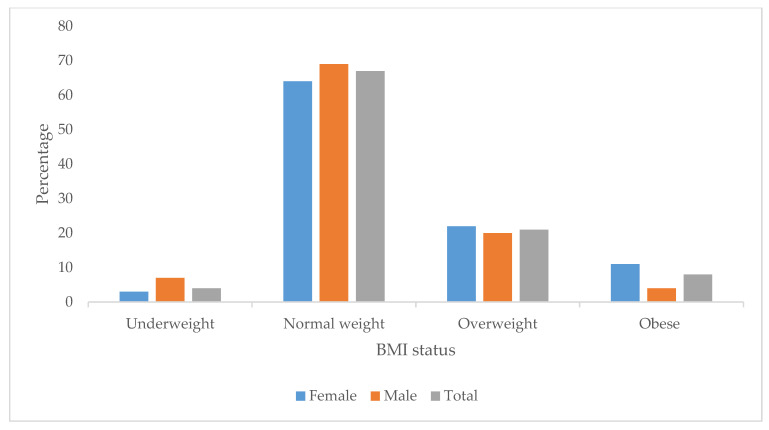
Body mass index of students (*n* = 363).

**Figure 2 ijerph-22-00936-f002:**
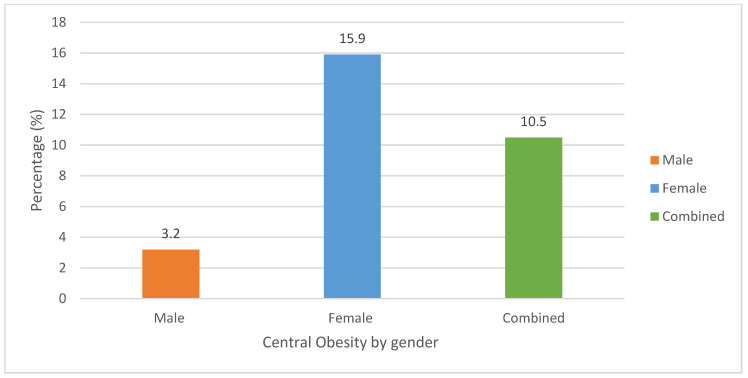
Prevalence of central obesity amongst students. Note: The cut-off points of waist circumference ≥ 88 cm for females and ≥102 cm for males were interpreted as central obesity.

**Table 1 ijerph-22-00936-t001:** Socio-demographic characteristics of students.

Characteristics	*n*	%
Age category:		
18–24	322	88.7
25–30	41	10.7
Gender:		
Males	155	42.7
Females	208	57.3
Ethnic group:		
Northern and Southern Sotho	72	19.8
Venda	154	42.4
Tsonga	78	21.5
Swati	40	11.0
Other ethnic groups	19	5.2
Academic level:		
First year	83	22.9
Second year	116	32.0
Third year	112	30.9
Honours	36	9.9
Masters	14	3.9
Doctoral	2	0.6
School:		
Agriculture	25	6.9
Education	67	18.5
Environmental science	45	12.4
Health Sciences	39	10.7
Human& social sciences	49	13.5
Law	17	4.7
Management sciences	63	17.4
Maths and natural sciences	58	16.0
Source of income:		
Bursary	294	81.0
Parents	36	9.9
Self-employed	6	1.7
Both parents and bursary	27	7.4

**Table 2 ijerph-22-00936-t002:** Dietary practices of the students (*n* = 363).

Food Items	Daily		Once Per Week		Twice Per Week		Three Times Per Week		Four Times or More		Not at All	
	*n*	(%)	*n*	(%)	*n*	(%)	*n*	(%)	*n*	(%)	*n*	(%)
Fast food or restaurant	20	5.5	153	42.1	42	11.6	15	4.1	21	5.8	112	30.9
Fruits and vegetables	79	21.8	75	20.7	54	14.9	63	17.4	79	21	13	3.6
Fish, chicken, eggs, lean meat	101	27.8	43	11.8	27	7.4	67	18.3	121	33.3	4	1.1
Beans	9	2.5	149	41	25	12.7	12	3.3	13	3.6	136	36.9
Processed food	31	8.5	103	28.4	70	19.3	37	10.2	22	6.1	100	27.5
Milk, mass, or yoghurt	66	18.2	92	25.3	63	17.4	39	10.7	82	14.3	51	14
Soft drinks, juice, and energised drinks	94	25	68	18.7	59	16.3	54	16.3	63	17.4	25	6.9

**Table 3 ijerph-22-00936-t003:** Relationship between dietary practices and body mass index status.

Variables	BMI Classification			*p*-Value
	Underweight	Normal	Overweight/obese	
Soft drinks, juice, and energised drinks	*n* (%)	*n* (%)	*n* (%)	0.006 **
Daily	10 (10.6)	61 (64.9)	23 (24.5)	
Once a week	4 (5.9)	39 (57.4)	25 (36.8)	
Twice a week	0 (0)	40 (67.8)	19 (32.2)	
Three and more per week	15 (4.1)	81 (69.8)	27 (23.3)	
Not at all	1 (3.8)	10 (38.5)	15 (57.7)	

Note: Chi-square (χ^2^) tests, significance levels: ** *p* < 0.01.

**Table 4 ijerph-22-00936-t004:** Relationship between consumption frequency per week WHR status.

Variables	WHR		*p*-Value
	Normal	At risk	
Frequency of eating in restaurants per week	*n* (%)	*n* (%)	
Daily	20 (100)	0 (0.0)	0.038 *
Once per week	143 (93.5)	10 (6.5)	
Twice and more	71 (91)	7 (9)	
Not at all	111 (99.1)	1 (0.9)	
Frequency of eating processed food per week			
Daily	31 (100)	0 (0.0)	0.007 **
Once per week	91 (88.3)	12 (11.7)	
Twice per week	68 (97.1)	2 (2.9)	
Three times and more per week	58 (98.3)	1 (1.7)	
Not at all	97 (97)	3 (3)	
Frequency of eating milk, mass, or yoghurt per week			
Daily	64 (97)	2 (3)	
Once per week	92 (100)	0 (0)	
Twice per week	59 (93.7)	4 (6.3)	0.013 *
Three time and more per week	81 (89)	10 (11)	
Not at all	49 (96.1)	2 (3.9)	
Frequancy of eating beans per week			
Daily	6 (66.7)	3 (33.3)	
Once per week	139 (93.3)	10 (6.7)	0.000 ***
Twice per week	46 (100)	0 (0)	
Three time and more per week	22 (91.7)	2 (8.3)	
Not at all	132 (97.8)	3 (2.2)	

Note: chi-square (χ^2^) tests, significance levels: * *p* < 0.05, ** *p* < 0.01, *** *p* < 0.001.

**Table 5 ijerph-22-00936-t005:** Median WHR values by dietary consumption frequency per week.

Dietary Factor	Not at All	Once Weekly	Twice Weekly	3+ Times Weekly	Daily	χ^2^	*p*-Value
Meals per day	-	-	0.79	0.78	0.78	3.06	0.217
Eating in Restaurants	0.77 ^a^	0.78 ^a^	0.80 ^b^	-	0.78 ^a,b^	15.54	0.001 **
Fruit intake	0.77	0.77	0.79	-	0.78	3.73	0.292
Protein foods	0.77 ^a,b^	0.75 ^a^	0.80 ^a,b^	0.79 ^b^	0.78 ^a,b^	12.23	0.016 *
Beans/legumes	0.78 ^a^	0.78 ^a^	0.80 ^a^	0.77 ^a^	0.83 ^b^	16.58	0.002 **
Processed foods	0.79	0.79	0.78	0.78	0.78	0.89	0.926
Dairy products	0.77	0.79	0.78	0.79	0.78	7.26	0.123
Sweetened beverages	0.81	0.80	0.78	0.79	0.78	6.13	0.190

Note: Different superscript letters (a, b) indicate significant differences between groups based on Dunn’s post hoc test with Bonferroni correction (*p* < 0.05). Significance levels: * *p* < 0.05, ** *p* < 0.01.

## Data Availability

The datasets generated and analysed in the current study are not publicly available online because the university has copyright. The datasets used during the current study are available from the corresponding author upon reasonable request.

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
