# Peer review of "Dietary Practices and Anthropometric Status of the Rural University Students in Limpopo Province, South Africa"

_ijerph, 2025, doi:10.3390/ijerph22060936_

Round 1

Reviewer 1 Report (Previous Reviewer 2)

Comments and Suggestions for Authors The article has improved, and I'm glad to know the authors have considered the issues raised by the reviewers. I believe the article can be accepted, but I would add the following as a note:
I would like to point out, among other limitations, that there are different parameters for assessing malnutrition and that the BMI should be complemented with other tools.  

Reviewer 2 Report (Previous Reviewer 1)

Comments and Suggestions for Authors

Dear Authors,

Thank you for your corrections to the manuscript. I have no further comments.

This manuscript is a resubmission of an earlier submission. The following is a list of the peer review reports and author responses from that submission.

Round 1

Reviewer 1 Report

Comments and Suggestions for Authors

Dear Authors,

Data concerning various age groups, including students, is of paramount importance in identifying factors contributing to the development of overweight and obesity. Information from different regions of the world, encompassing diverse social groups, provides a broader perspective on this issue. Although this topic has been widely discussed, the authors selected the study population based on its unique characteristics.

Introduction

  • The introduction has been appropriately structured; the cited literature is sufficient to highlight the research problem. The study's objective has been clearly outlined.

Materials and Methods

  • The statistical methods employed are relatively basic. It is worth considering whether non-parametric tests could be applied for BMI, such as ANOVA in the case of normal distribution or the Kruskal-Wallis/Mann-Whitney U test in the absence of normality.
  • Have the authors considered using the waist-to-height ratio? Incorporating this metric might provide additional insights into the findings.

Results

  • The results have been clearly and precisely described. It would be beneficial to include the outcomes of non-parametric tests and, if applicable, results related to the waist-to-height ratio.

Discussion

  • The discussion has been well-developed, with the authors presenting numerous comparisons with findings from other studies. After incorporating the reviewer’s suggestions, a more comprehensive discussion can be expected.

Reviewer 2 Report

Comments and Suggestions for Authors

Review the journal's guidelines because the line spacing may be incorrect.
Introduction:
- Well-structured but too long; it is recommended to shorten the text. References are missing on lines 79-82.
- Although no data have been published in the same population, there are numerous studies that address similar objectives; the need for this study should be better justified.
- The "Conceptual framework" section should be removed or should be part of the introduction as a justification.

Methodology
- Repetition of the objective should be removed (lines 108-110).
- Indicate whether the study was approved by the ethics committee and whether the participants were informed and gave their consent to participate in the study.

- Reference 14 is incorrect; it should be 15. Furthermore, reference 15 is outdated, and the correct reference is "Esparza-Ros F, Vaquero-Cristóbal R, Marfell-Jones M. International Standards for Anthropometric Assessment. International Society for Advancement of Kinanthropometry; 2019." The authors should review all citations/references in the manuscript because they are incorrect.
- In the Dietary Practices section, the type of questionnaire used, its validation, and use for the population to which it was administered are unclear. The reference used is a general WHO document. Furthermore, the questionnaire includes a short list of food groups that do not allow us to determine whether the diet is adequate, healthy, etc. For example, it includes "Fish, Chicken, Eggs, Lean Meat" in the same group.

Discussion
- The last paragraph of the discussion should address the study's limitations and strengths.

Round 2

Reviewer 2 Report

Comments and Suggestions for Authors

The manuscript has been improved taking into account the reviewers' suggestions, but:
- Generally, an introduction does not include a figure; one could be briefly described if necessary.
- The new introduction lacks bibliographic references. Furthermore, the results do not show how the authors incorporate malnutrition into their presentation of results.
- Information from the ethics committee should be included in the "Study design and setting" section.
- The "Dietary practices" section should indicate what types of food groups the questionnaire includes.

Author Response

See the attached rebuttal letter 
